# Evolution, systematics and historical biogeography of sand flies of the subgenus *Paraphlebotomus* (Diptera, Psychodidae, *Phlebotomus*) inferred using restriction-site associated DNA markers

Astrid Cruaud[1], Véronique Lehrter[2], Guenaëlle Genson[1], Jean-Yves Rasplus[1‡], Jérôme Depaquit[2‡*]

1 CBGP, INRAE, CIRAD, IRD, Montpellier SupAgro, Montpellier, Univ Montpellier, France, 2 Université de Reims Champagne Ardenne, ESCAPE EA7510, USC ANSES VECPAR, SFR Cap Santé, UFR de Pharmacie, Reims, France

☯ These authors contributed equally to this work.
‡ These authors are joint senior authors on this work.
* jerome.depaquit@univ-reims.fr

**Data Availability Statement:** Fastq reads for analysed samples are available as a NCBI

## Abstract

Phlebotomine sand flies are the main natural vectors of *Leishmania*, which cause visceral and tegumentary tropical diseases worldwide. However, their taxonomy and evolutionary history remain poorly studied. Indeed, as for many human disease vectors, their small size is a challenge for morphological and molecular works. Here, we successfully amplified unbiased copies of whole genome to sequence thousands of restriction-site associated DNA (RAD) markers from single specimens of phlebotomines. RAD markers were used to infer a fully resolved phylogeny of the subgenus *Paraphlebotomus* (11 species + 5 outgroups, 32 specimens). The subgenus was not recovered as monophyletic and we describe a new subgenus *Artemievus* subg. nov. Depaquit for *Phlebotomus alexandri*. We also confirm the validity of *Ph. riouxi* which is reinstated as valid species. Our analyses suggest that *Paraphlebotomus sensu nov.* originated *ca* 12.9–8.5 Ma and was possibly largely distributed from peri-Mediterranean to Irano-Turanian regions. Its biogeographical history can be summarized into three phases: i) a first split between *Ph. riouxi* + *Ph. chabaudi* and other species that may have resulted from the rise of the Saharan belt ca 8.5 Ma; ii) a Messinian vicariant event (7.3–5.3 Ma) during which the prolonged drought could have resulted in the divergence of main lineages; iii) a recent radiation event (3–2 Ma) that correspond to cycles of wet and dry periods in the Middle East and the East African subregions during the Pleistocene. Interestingly these cycles are also hypothetical drivers of the diversification of rodents, in the burrows of which *Paraphlebotomus* larvae develop. By meeting the challenge of sequencing pangenomics markers from single, minute phlebotomines, this work opens new avenues for improving our understanding of the epidemiology of leishmaniases and possibly other human diseases transmitted by arthropod vectors.

Sequence Read Archive (PRJNA725143). COI sequences were uploaded in Genbank IDs #MZ049642-MZ049672. Data sets and trees have been uploaded on Zenodo (https://doi.org/10.5281/zenodo.4721254).

**Funding:** The authors received no specific funding for this work.

**Competing interests:** The authors have declared that no competing interests exist

## Author summary

Phlebotomine sand flies are tiny insects transmitting unicellular parasites called *Leishmania* worldwide. They are bad flyers and are known to be a group dispersing according to historical events. No invasive species are known within Phlebotomine sandflies. Their taxonomy includes about 40 genera. In the Old World, the subgenus *Paraphlebotomus* includes the main vectors of *Leishmania tropica*, the agent of a cutaneous leishmaniasis mostly in North Africa visceral and the Middle West. The goal of this study was to explore the phylogeny and the biogeographical history of this subgenus using a new generation sequencing technique. We successfully amplified DNA copies from the selected DNA extracts, some of them being more than 20 years old. We fully resolved the phylogeny of the subgenus *Paraphlebotomus* who was not recovered as monophyletic, meaning it does not include all the ascendants of last common ancestor. Consequently, the taxonomy was revised with the creation of a new subgenus. We also suggest that *Paraphlebotomus* originated about 10 million years ago. After a first split isolating North African species, the Messinian crisis caused by the aridification of the Tethys (the ancestor of the Mediterranean Sea) occurring 7 million years ago allowed the divergence of the main lineages including that of *Phlebotomus sergenti* the most important vector of the group. Most recent radiation events occurred during the Pleistocene. The historical biogeography of these Phlebotomine sandflies are closely related to that of the *Leishmania* they transmit.

## Introduction

Phlebotomine sand flies (Diptera, Psychodidae) are the main natural vectors of *Leishmania*, a genus of protozoan parasites which cause visceral and tegumentary neglected tropical diseases worldwide. Of the approximately 1,000 species of sand flies, about 70 are vectors of *Leishmania* [1,2]. Leishmaniases are endemic in nearly 100 countries, mainly in Africa, Asia and Latin America [3]. The estimated annual incidence of cutaneous leishmaniases in humans is 0.7 to 1 million cases and 50,000 to 90,000 cases of visceral leishmaniases were reported in 2017 [4]. However, most leishmaniases are zoonotic diseases affecting wild or domestic animals. Canine leishmaniasis is for instance a serious veterinary problem in many countries [14].

As for any vector-borne disease, an accurate systematics of the vectors is critical for a better understanding of the epidemiology of diseases. However, although phlebotomine sand flies are the subject of many research programs, their taxonomy and evolutionary history remain poorly studied [5], which limits our progress to reduce the incidence of leishmaniases.

Phlebotomine are minute flies (2–3 mm), which makes their identification to species difficult, especially for non-specialists. Diagnostic characters are almost exclusively located on the head and genitalia of both sexes and their observation require that specimens are slide mounted, which further complicates taxonomic work. The small size of the phlebotomine sand flies is also a technical challenge for molecular work, which may explain why only few molecular studies were conducted so far. Because species complexes may exist, the sequencing of a pool of specimens is hazardous and DNA extracted from single specimens should be used for molecular work. In addition, as dissection of the head and genitalia is required for identification, DNA extraction must be performed only on remains of the body. As a result, DNA yield is low, which drastically limits the number of genes that can be amplified and sequenced from a single specimen using PCR and Sanger approaches. Thus, the amount of genetic

information that can be analysed to infer robust relationships and highlight possible species complexes is low.

In the Old World, most vectors of *Leishmania* belong to the genus *Phlebotomus* [5], and some are included in the genus *Sergentomyia* [6]. Eleven subgenera have been described in the genus *Phlebotomus* [7] among which, four are involved in the transmission of *Leishmania*, namely *Larroussius*, *Phlebotomus*, *Euphlebotomus* and *Paraphlebotomus*, and two others are suspected or local vectors (*Adlerius* and *Synphlebotomus*) (Table 1).

Based on morphological characters [8], *Paraphlebotomus*, *Phlebotomus* and *Synphlebotomus* are closely related (presence of a basal lobe on gonocoxite, short aedeagal or spermathecal ducts, short parameral sheath, annealed spermathecae without neck), while *Larroussius* (lack of basal lobe on gonocoxite, long aedeagal or spermathecal ducts, long parameral sheath, annealed spermathecae with neck) appears to be part of another evolutionary lineage also including *Adlerius*, *Euphlebotomus*, *Anaphlebotomus* or *Transphlebotomus*. Two studies based on morphological characters consider these two groups of subgenera as separate clades [9,10]. Studies based on ribosomal DNA, highlighted that the subgenera *Paraphlebotomus* and *Phlebotomus* belong to the same clade [11,12] but no molecular study has been published so far that included members of these related subgenera.

Here, we focus on the subgenus *Paraphlebotomus* that currently includes 14 species (Table 1), four of which being demonstrated vectors of *Leishmania* spp. The most important vector is undoubtfully *Ph. sergenti*, which is the main vector of *L. tropica*, the etiological agent of an anthroponotic cutaneous leishmaniasis occurring in a large area from Pakistan to Morocco [13,14]. However, *L. tropica* can be locally transmitted by other vectors belonging to

**Table 1. Subgenera of *Phlebotomus* sand flies involved in *Leishmania* transmission.**

| Subgenus | Species | Main *Leishmania* species transmitted (suspected or **proven**) | References | Included in the study |
|---|---|---|---|---|
| *Adlerius* | *Ph. (Ad.) simici* | *L. infantum* | [1] | x |
| | *Ph. (Ad.) arabicus* | *L. tropica* | [17] | |
| *Euphlebotomus* | *Ph. (Eu.) argentipes* | **L. donovani** | [1, 94] | / |
| *Larroussius* | *Ph. (La.) neglectus* | **L. infantum** | [95] | x |
| *Paraphlebotomus* | *Ph. (Para.) alexandri* | *L. donovani* | [1, 96] | x |
| | | **L. infantum** | [97] | |
| | *Ph. (Para.) andrejevi* | *L. major* | [98] | / |
| | *Ph. (Para.) caucasicus* | **L. major** | [99] | x |
| | *Ph. (Para.) chabaudi* | / | / | x |
| | *Ph. (Para.) gemetchi* | / | / | / |
| | *Ph. (Para.) jacusieli* | / | / | x |
| | *Ph. (Para.) kazeruni* | *L. major* | [1, 100] | x |
| | *Ph. (Para.) mireillae* | / | / | x |
| | *Ph. (Para.) mongolensis* | / | / | x |
| | *Ph. (Para.) nuri* | / | / | / |
| | *Ph. (Para.) riouxi* | / | / | x |
| | *Ph. (Para.) saevus* | **L. tropica** | [101] | x |
| | *Ph. (Para.) sergenti* | **L. tropica** | [13, 14, 101] | x |
| | | **L. aethiopica** | [101] | |
| | *Ph. (Para.) similis* | / | / | x |
| *Phlebotomus* | *Ph. (Ph.) duboscqi* | **L. major** | [102] | x |
| | *Ph. (Ph.) papatasi* | **L. major** | [103] | x |
| *Synphlebotomus* | *Ph. (Sy.) ansarii* | *L. major* | [104] | x |

subgenera *Larroussius* in Kenya [15,16] and *Adlerius* in Israel/Palestine [17]. So far, only three molecular studies have included enough species of *Paraphlebotomus* to provide preliminary hypotheses about its systematics. Depaquit et al. [18] have conducted the only molecular study that attempted to resolve phylogenetic relationships within *Paraphlebotomus* using nine ingroup and two outgroup species. However, a single marker (*Internal* transcribed spacer 2, ITS2) was used and the resolution was low. Besides, we cannot exclude that possible paralogs of ITS2 may have biased results. A second study [19], whose goal was to describe the diversity of sand flies in military camps in Afghanistan using mitochondrial Cytochrome b gene (*Cytb*), generated data for five species of *Paraphlebotomus*. In this study, *Paraphlebotomus* appeared as a polyphyletic assemblage. *Ph. alexandri* grouped with species from the subgenus *Phlebotomus* (*Ph. papatasi* and *Ph. bergeroti)* with poor support. Finally, Grace-Lema et al. [20] used a patchy matrix of 7 genes (65% missing data, *ca* 45% of the species with more than 85% missing data) to infer a phylogeny of phlebotomine sand flies (113 terminals). However, as for the rest of the tree, relationships between the four species of *Paraphlebotomus* were poorly resolved. Thus, the systematics of the subgenus is still unclear.

Most species of *Paraphlebotomus* have restricted distribution (Fig 1; see also [21]). *Ph. mireillae* is found in Kenya and southern Ethiopia [22]; *Ph. saevus* occurs in Arabian Peninsula and in the Horn of Africa; *Ph. nuri* is a Middle-East endemic and *Ph. gemetchi* is reported from Ethiopia [23]. *Ph. chabaudi* and *Ph. riouxi* are Ibero-Maghrebian and *Ph. similis* occurs in the north-east of the Mediterranean basin, with recent records from central Turkey [24], Iran [25] and Ukraine [26]. *Ph. jacusieli* is distributed from eastern Mediterranean to eastern Iran [27]. *Ph. caucasicus*, *Ph. andrejevi* and *Ph. mongolensis* are located in central Asia or Middle-east, but the range of *Ph. mongolensis* extents to East China sea (Suizhong, Liaoning province) [28]. To the opposite, a few species have extremely large distribution. *Ph. sergenti* and *Ph. alexandri* occur from Europe and North Africa to Middle and Far East. *Ph. kazeruni* is almost always a rare species, mainly recorded from Iran, but recent records in Turkey [29], southern Algeria [30] and Pakistan [31,32] have widened its known range. As suggested by Depaquit et al. [18], the ancestor of the subgenus *Paraphlebotomus* may have originated in Eastern Africa or in the Arabic peninsula, however this needs to be formally tested.

With technical advances, it is now possible to generate unbiased copies of whole genomes (Whole Genome Amplification, WGA) to circumvent low DNA amount issues [33–35]. Although this technic is still rarely used for such purpose, it opens new avenues for the sequencing of pangenomic markers to resolve relationships among tiny species. Here we used WGA and RAD (Restriction site Associated DNA) sequencing [36] of single specimens of phlebotomine sand flies to infer a robust phylogeny of the subgenus *Paraphlebotomus* and five outgroups. We also performed divergence time estimates and probabilistic inference of historical biogeography of the subgenus.

## Materials and methods

### Taxonomic sampling

Twenty-three ingroup specimens representing 11 species (78% of the known diversity) were selected as ingroup. In average, one or two specimens per species were analysed. We failed to collect two rare species (*Ph. nuri* and *Ph. gemetchi*) and to process old DNA extracts of *Ph. andrejevi*. For demonstrated vectors, four to five specimens collected in several locations were included. Nine outgroup specimens (five species) were chosen in the subgenera *Phlebotomus*, *Synphlebotomus*, *Larroussius* and *Adlerius*. Sampling was carried out during the last twenty years, from a large area extending from Senegal to India (west-east axis) and from Italy to Kenya (north-south axis) (Fig 1 and S1 Table). Samples were collected using CDC trap, sticky

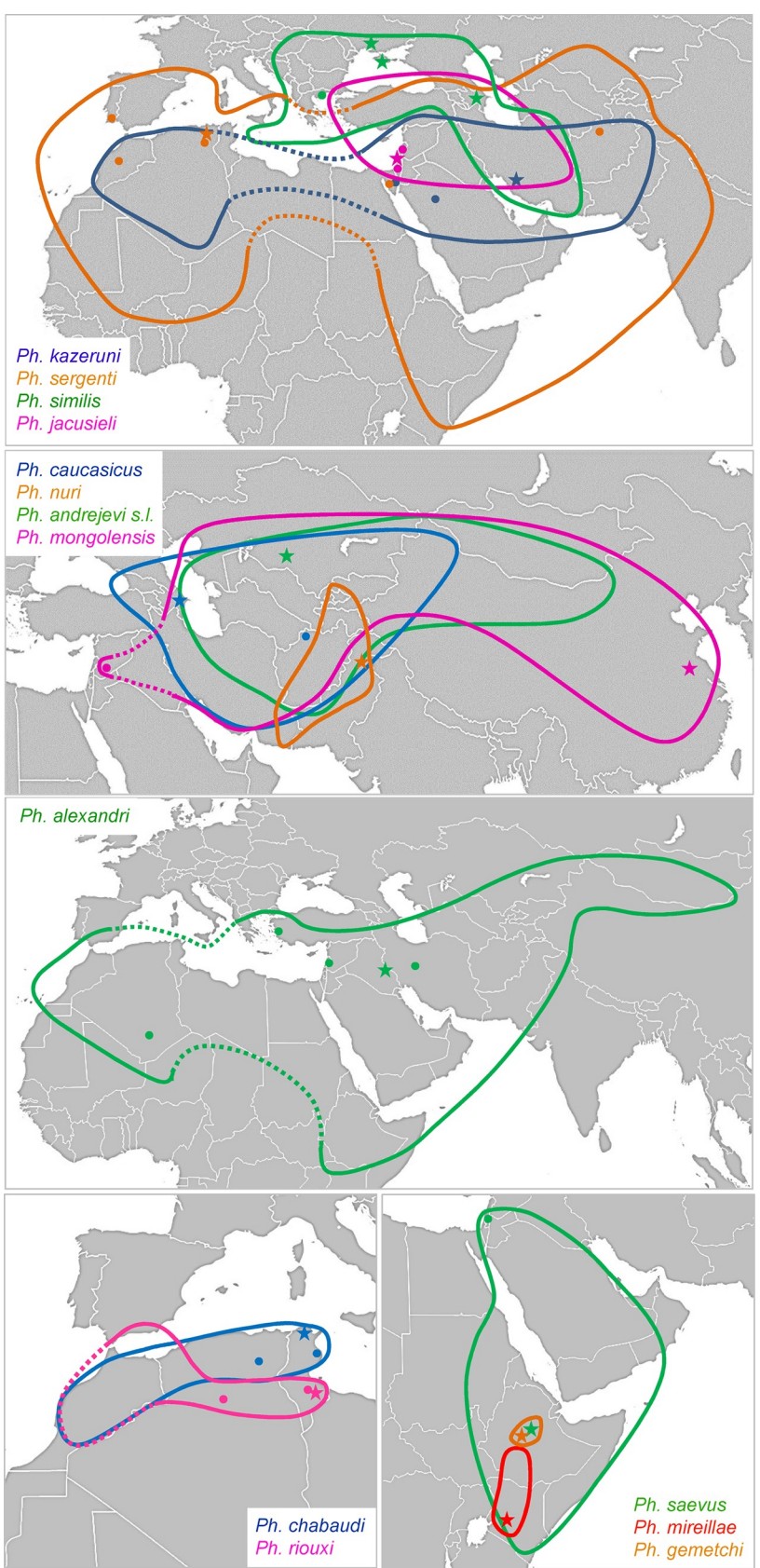

**Fig 1. Distribution maps and sampling sites for the different species of *Paraphlebotomus sensu* Lewis [7] and Depaquit [54].** Ranges are delimited by color lines. Lines are dotted when the presence is expected but has never been confirmed (lack of sampling, doubtful taxonomic status). Sampling sites and type localities are respectively represented by dots and stars. The background map was in the public domain and has been downloaded from https://commons. wikimedia.org/wiki/File:BlankMap-World.svg.

trap, or came from colonies reared at lab. Specimens were preserved in 95% ethanol at room temperature or at -20˚C until dissection. Some samples were dried freeze at -80˚C or in liquid nitrogen for longer conservation. After thawing, specimens were dissected in 95% ethanol. For morphological identification under stereomicroscope, head and genitalia were mounted in chloral gum or in Euparal between slide and cover slide after clarification in boiling Marc André solution, washing and dehydration. The rest of the body was dried freeze at -20˚C for molecular biology analyses. Identification to species was performed by JD and vouchers (slides + body remains after DNA extractions) have been deposited in the University of Reims Champagne-Ardenne, collection of J. Depaquit.

## DNA extraction and whole genome amplification

DNA extractions were performed with either CTAB/Chloroform-Isoamyl Alcohol or extraction kits. For the CTAB/Chloroform-Isoamyl Alcohol extraction, samples were crushed and incubated at 60˚C in a solution containing CTAB buffer [37]. Proteins were eliminated by the addition of chloroform-Isoamyl Alcohol. RNase (0.5 units) was added to the aqueous phase, which was then incubated at 37˚C for 30 min to remove RNA. Total genomic DNA was precipitated by the addition of two-thirds of the total volume of isopropanol and stored at 4˚C overnight. After centrifugation, the pellet was washed, dried and suspended in 50 μL of ultrapure water. Genomic DNA was also extracted using the QIAamp DNA Mini Kit (Qiagen) following the manufacturer's instructions. DNA was eluted using 100 μL to 160 μL of elution buffer.

DNA quantity was measured using the Qubit double-stranded DNA High Sensitivity Assay kit (Thermofisher Scientific) following the manufacturer's instructions. DNA extracts were then subjected to Whole Genome Amplification using the GenomiPhi V2 DNA Amplification kit (GE Healthcare). Ethanol precipitation of DNA was performed prior to WGA to fit with the recommendations of the GenomiPhi protocol (1μL DNA input at 10 ng/uL). 1/10 volume of sodium acetate 3 M pH 5.2 was added to DNA. Then, 2 volumes of cooled absolute ethanol were added to the mix. The mix was incubated at -20˚C overnight. The pellet of DNA extract obtained by centrifugation (30 min, 13 000 rpm, 4˚C) was washed by 500 μL of cooled 70% ethanol. After a new centrifugation (15 min, 13 000 rpm, 4˚C), the pellet was dried at room temperature, and suspended in 4 μL of sterile molecular biology ultrapure water. Whole Genome Amplification (WGA) was then performed on 1 to 2 μL of concentrate DNA following the manufacturer protocol.

## DNA barcoding

The standard *COI* barcode fragment was sequenced for all specimens to obtain calibrations for divergence time estimates (see below). For each sample, the cocktail of M13 tailed primers and the PCR conditions described in Germain et al. [38] were used on 4μL of DNA obtained with WGA. Unpurified PCR products were sent to Eurofins Genomics for Sanger sequencing. Forward and reverse strands were assembled using Geneious 10.2.2 [39]. Consensuses were aligned with MAFFT-7.313 (linsi option) [40] and translated to amino acids using MEGA 7

[41] to detect frame-shift mutations and premature stop codons, which may indicate the presence of pseudogenes. Sequences were merged with all barcodes available in BOLD [42] for the genus *Phlebotomus* (last access December 4, 2019) and IQ-TREE v1.6.7 [43] was used to infer a maximum likelihood phylogenetic tree. The best-fit evolutionary model was selected with ModelFinder as implemented in IQ-TREE [44] and branch supports were assessed with ultrafast bootstrap [45] and SH-aLRT test [46] (1000 replicates).

## Preparation and sequencing of RAD library

In silico digestion of the genome of *Ph. papatasi* available on GenBank (AJVK00000000.1) was performed using the recognition site of the *PstI* enzyme to determine the number of expected cut-sites and calculate the maximum number of samples to be sequenced on a single lane of an Illumina HiSeq 2500. 51,712 cut sites (103,424 RAD markers) were inferred. Considering that *ca* 150,000,000 passing filter reads would be usable to construct RAD loci according to our experience (after removal of reads with mutation in barcodes /restriction sites and PCR duplicates), 48 samples were included in the library to get a sequencing depth per marker of ca 30X (only 32 samples were analysed for this study, the 16 others were sequenced for another purpose). Library construction followed by Baird et al. [36] and Etter et al. [47] with modifications detailed in Cruaud et al. [48]. The quantity of P1 adapters to be added to saturate restriction sites (result = 2 μL) as well as the optimal time for DNA sonication in a Covaris S220 ultrasonicator (result = 70s) that are both specific to the studied group were evaluated in a preliminary experiment. After tagging with barcoded P1 adapters and prior to sonication, specimens were pooled eight by eight (six pools were thus obtained and tagged with barcoded P2 adapters). 2*125nt paired-end sequencing of the library was performed at MGX-Montpellier GenomiX on one lane of an Illumina HiSeq 2500 flow cell.

## RAD data cleaning and phylogenetic inference

The quality of the raw data was assessed using FastQC (Babraham Institute, Bioinformatics Group http://www.bioinformatics.bbsrc.ac.uk/projects/fastqc). Raw data were processed using the Perl wrapper RADIS [49] that relies on Stacks [50,51] for demultiplexing of data, removing PCR duplicates and building individual and catalog loci. For data cleaning (RADIS_step1_data_cleaning.pl), parameters in the RADIS.cfg files were set as follows (radis_readlen = 125; radis_nttrim_read1_5p = 5 to remove the complementary overhang of the restriction site; radis_nttrim_read1_3p = 0; radis_nttrim_read2_5p = 0, radis_nttrim_read2_3p = 0).

Individual RAD loci were built using m (minimum depth of coverage to create a stack) = 3 and, M (maximum distance allowed between stacks) = 2; with removal (r) and deleveraging (d) algorithms enabled. Six data sets were then built to test a possible impact of analytical parameters on our phylogenetic analyses. Catalogs of loci were built with n (number of mismatches allowed between sample tags when generating the catalog) set to 6, 8 or 10. Loci / samples selection was performed as follows: only samples with more than 100 RAD loci in the catalog were kept (radis_nloci_min), only locus for which at least either 50% or 75% of the samples had sequences were kept (radis_nsample_min) and loci for which at least one sample had 3 sequences or more were removed to discard possible paralogs (radis_npbloci_cutoff). Phylogenetic inferences were performed using RAxML 8.2.9 (multi-threaded version with AVX extensions) [52] and a rapid bootstrap search with 100 replicates, followed by a thorough ML search (GTR GAMMA), without partitioning of the data set. Trees were visualized and annotated with TreeGraph 2.13.0-748beta [53]. Percentage of missing data and percentage of identity between sequence pairs were calculated using Geneious 10.2.2 [39].

## Species distribution, divergence time estimation and ancestral range estimation

Species distribution for subgenus *Paraphlebotomus* were estimated from maps of two precedent studies [7,54] and a review of recent papers (see results for an exhaustive list). There is no fossil record of *Phlebotomus* and paleogeographic events that could be used as calibrations are unknown. Consequently, we used calibration priors based on mitochondrial DNA. We estimated minimum and maximum pairwise sequence divergences among *COI* sequences of sampled belonging to the two species pairs recovered both in the RAD and *COI* trees (i.e. *Ph. chabaudi-Ph. riouxi* and *Ph. papatasi-Ph. duboscqi*) using MEGA7 (evolutionary model = K2P). Divergences were then divided by the *COI* clock rate for insects estimated by Papadopoulou et al. [55] (i.e 3.54 ± 0.38% divergence per Myr) to obtain minimum and maximum ages for the split between the two species of each pair following Godefroid et al. [56]. A timetree was built with the RelTime method [57] as implemented in MEGA X [58] using the RelTime-ML option, a GTR+G evolutionary model (with 4 discrete Gamma categories as for the RAxML analysis) and all nucleotide sites (including gaps and missing data). The RAxML RAD tree obtained with the cstacks n = 10, radis_nsample_min = 50% of the samples (Table 2; ie the tree obtained with the largest data set) was used as input and *Ph.* (*Adlerius*) *simici* and *Ph.* (*Larroussius*) *neglectus* were considered as outgroups. Uniform distributions were used as calibration densities.

We inferred the biogeographic history of the subgenus *Paraphlebotomus* using the R package BioGeoBEARS 1.1.1 [59]. The chronogram built with RelTime was used as input but only one specimen per species was included in the analysis and outgroups (including *Ph. alexandri*) were removed to avoid artefacts (taxa were pruned from the chronogram with the R package ape [60]). We compared the fit of the models DEC, BAYAREALIKE, and DIVALIKE with AICc but did not consider the jump parameter for founder events (+J) following recent criticisms [61]. Geographic ranges of the species were coded using six geographical subregions of the West Palearctic: A: West Mediterranean; B: Saharo-Arabian; C: Sudanian; D: Somalian; E: East Mediterranean; F: Irano-Turanian. The maximum number of areas species may occupy was set to 6. To consider the main geologic events that occurred during the last 15 Myr in the peri-mediterranean area, we defined two time periods with different dispersal rate scalers: 1) from the Mid Miocene climatic optimum (ca 15 Ma) to the end of the Messinian crisis (5.3 Ma); 2) from the Zanclean flood (5.3 Ma) to present. The matrix of geographical ranges for the

**Table 2. Properties of the RAD data sets obtained with different parameters combinations.** n = number of mismatches allowed when merging individual loci in cstacks, radis_nsample_min = minimum number of samples that should have sequences for a RAD locus to be included in the analysis. In all data sets, loci for which at least one sample had three or more sequences (*i.e.* possible paralogs) were removed (radis_npbloci_cutoff = 3).

| Combination of parameters | # RAD markers | # bases in alignment | % missing data | % variable sites | % parsimony informative sites | % GC | RAxML tree |
|---|---|---|---|---|---|---|---|
| n = 6, radis_nsample_min = 50% of the samples | 5,050 | 580,750 | 34.6 | 12.0 | 8.40 | 45.4 | S2A Fig |
| n = 6, radis_nsample_min = 75% of the samples | 1,423 | 163,645 | 16.7 | 11.5 | 8.20 | 45.9 | S2B Fig |
| n = 8, radis_nsample_min = 50% of the samples | 7,194 | 827,310 | 34.1 | 14.7 | 10.6 | 45.8 | S2C Fig |
| n = 8, radis_nsample_min = 75% of the samples | 2,086 | 239,890 | 16.6 | 14.4 | 10.6 | 46.4 | S2D Fig |
| n = 10, radis_nsample_min = 50% of the samples | 8,906 | 1,024,190 | 33.1 | 16.9 | 12.4 | 46.1 | S2E Fig |
| n = 10, radis_nsample_min = 75% of the samples | 2,808 | 322,920 | 16.7 | 16.9 | 12.7 | 46.7 | S2F Fig |

different species as well as the dispersal multipliers for each time period are available as S1 Text.

### Computational resources

Analyses were performed on the Genotoul Cluster (INRAE, France, Toulouse).

## Results

### DNA barcoding and calibrations

*COI* sequences were successfully obtained from all samples but one (*Ph. saevus* for which all DNA was included in the RAD library). The maximum likelihood tree confirmed that *COI* is not informative enough to solve relationships among species of phlebotomine sand flies (S1 Fig). To avoid issues due to mitochondrial introgression and putative paralogs (heteroplasms, coding pseudogenes) that could be contained in the *COI* data set, only splits between highly supported species pairs that were recovered both in the RAD (S2 Fig) and the *COI* trees were used to calibrate our dating analysis, namely *Ph. chabaudi-Ph. riouxi* and *Ph. papatasi-Ph. duboscqi*. Minimum and maximum interspecific K2P distances resulted in minimum and maximum ages of 2.42–3.64 Myr for the *Ph. chabaudi-Ph. riouxi* split (rounded to 2.0–4.0 Myr for the RelTime analysis) and 2.83–4.27 Myr for the *Ph. papatasi-Ph. duboscqi* split (rounded to 2.0–5.0 Myr for the RelTime analysis).

### RAD data sets and phylogenetic inferences

In average, 97,054 RAD loci were obtained per specimen (ustacks, S1 Table). Depending on the value of the parameters n (number of mismatches allowed when merging individual loci) and radis_nsample_min (minimum number of samples that should have sequences for a RAD locus to be included in the analysis) data sets included 1,423 to 8,906 RAD loci of 115 bp each and contained only 16.6–34.6% missing data (Table 2).

All RAxML analyses yielded the same fully resolved tree topology (Figs 2 and S2). The sub-genus *Paraphlebotomus* was paraphyletic. *Ph. alexandri* was recovered sister to the rest of *Paraphlebotomus* species and a clade consisting of members of the *Phlebotomus* (*Ph. papatasi* and *Ph. duboscqi*) and *Synphlebotomus* (*Ph. ansarii*) subgenera. The remaining *Paraphlebotomus* species clustered into a strongly supported clade in which relationships between species were fully resolved.

### Species validity and intraspecific variability

When multiple conspecific specimens were analysed, they formed well-supported clades that were deeply separated from their nearest relative, suggesting that morphology alone mostly enables accurate identification of species. This was the case for the analysed specimens belonging to *Ph. chabaudi* and *Ph. riouxi* with an average pairwise percentage dissimilarity between specimens of the two clades of 1.7% for the n = 10, radis_nsample_min = 50% data set (i.e. on 1,024,190 nt). Significant genetic divergence was observed between individuals of species with large distribution ranges (*Ph. alexandri*–up to 1.6% dissimilarity and *Ph. sergenti*–up to 1.2%).

### Divergence-time estimates

RelTime analysis inferred that the split between *Ph. alexandri* and all other lineages has occurred about 15 Ma (14.7 Ma, confidence interval–CI = 10.6–20.2) and that the subgenus *Paraphlebotomus* (excluding *Ph. alexandri*) originated in the Middle—late Miocene between 12.9 (CI = 9.3–17.9; most ancient ancestor) and 8.5 Ma (CI = 5.7–12.8; most recent common

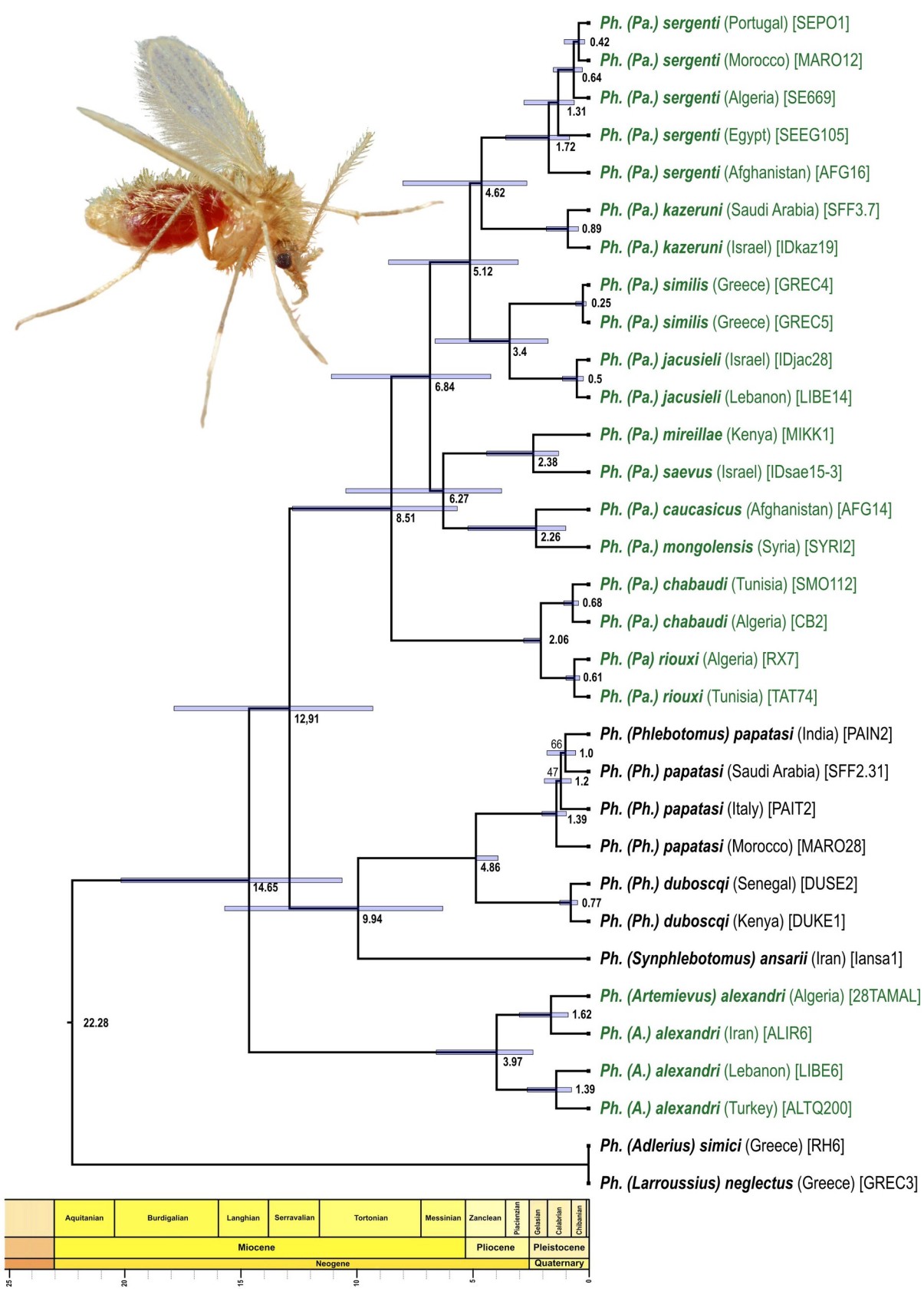

**Fig 2. RelTime timetree.** Bootstrap supports lower than 100 are reported at nodes (100 replicates, RAD data set used: cstacks n = 10, radis_nsample_min = 50% see Table 2 and S2 Fig). Species belonging to subgenus *Paraphlebotomus sensu* Seccombe et coll., 1993 are shown in green but terminals are annotated following the new classification proposed in this study with *Ph. alexandri* belonging to *Artemievus* **subg. nov.** Depaquit. Divergence times are not displayed in the outgroup (*Ph.* (*Adlerius*) *simici* and *Ph.* (*Larroussius*) *neglectus*) because RelTime uses evolutionary rates from the ingroup to calculate divergence times and does not assume that evolutionary rates in the ingroup clade apply to the outgroups. The image shows *Ph. papatasi* and has been modified from the image #10275 of the Centers for Disease Control and Prevention's Public Health Image Library (PHIL: https://phil.cdc.gov/; Content provider: CDC/ Frank Collins. Photo credit: James Gathany; last access January 22, 2021) that is in the public domain and thus free of any copyright restrictions.

ancestor MRCA) (Fig 2). Within *Paraphlebotomus*, the "*Ph. chabaudi / Ph. riouxi*" lineage split from all other *Paraphlebotomus* sand flies about 8.5 Ma by the end of Miocene. All deep splits between geographical lineages occurred between 6.8 and 5.1 Ma, although with relatively large confidence intervals. Finally, most splits between sister species occurred during the Pliocene between 4.6 Ma (*Ph. sergenti / Ph. kazeruni*) and 2.3 Ma (*Ph caucasicus / Ph. mongolensis*). The main split within *Ph. alexandri* also occurred during this time period (ca 4.0 Ma).

## Biogeography

AICc values show that the DEC and DIVALIKE models have an equally significant fit (S3 Text). The ancestral range of *Paraphlebotomus* is equivocal with Irano-Turanian + West Mediterranean subregions as the most likely range (Fig 3 and S4 Text). Each of the five species pairs (namely *riouxi-chabaudi; mongolensis-caucasicus; saevus-mireillae; jacusieli-similis* and *kazeruni-sergenti*) originated in a different subregion of the West-Palearctic (respectively West-Mediterranean, Irano-Turanian, Somalian, East-Mediterranean and Saharo-Arabian), suggesting an origin by vicariant fragmentation. The only difference between the DEC and DIVALIKE models is the most likely range for the MRCA of the clade *mongolensis-caucasicus; saevus-mireillae; jacusieli-similis* and *kazeruni-sergenti*, which is Somalian + Irano-Turanian subregions for DEC and Irano-Turanian only for DIVALIKE (S4 Text).

## Discussion

### Description of a new subgenus of *Phlebotomus* Rondani & Berté

*Ph. alexandri* did not cluster with other species of *Paraphlebotomus*, which questions its current classification. An early divergence of *Ph. alexandri* [18] as well as the paraphyly of the subgenus *Paraphlebotomus* have already been acknowledged [19] and is confirmed here with high statistical support. Based on morphology, *Ph. alexandri* greatly differs from other species of *Paraphlebotomus* [54]. The first flagellomere (f1 = antennal segment AIII) is extremely short and the basal lobe of the coxite is wider than long. The pharyngeal armature is poorly extended and composed by a network of conspicuous scaly teeth. These apomorphies lead us to propose a new subgenus for *Ph. alexandri*.

Artemievus *subg. nov.* Depaquit. Type-species: *Phlebotomus alexandri* Sinton, 1928.

*Artemievus* subg. nov. is defined by i) a short (<170μm) flagellomere 1 (= A III), shorter than labrum and shorter than the combined length of flagellomeres 2 and 3 (= AIV + AV) in both genders, ii) in females, a rectangular pharyngeal armature occupying less than a quarter of the pharynx, spermathecae with seven to 10 rings, ascoids reaching the next articulation, iii) in males, basal lobe of gonocoxite not incurved, style with four spines, the distal one carried by a long process, paramere simple and spoon-like and a narrow sperm pump (Figs 4 and 5).

This subgenus includes only *Ph.* (*A.*) *alexandri*, a widespread and polymorphic species extending from Morocco to Mongolia and from Ethiopia to southern Europe (Fig 1).

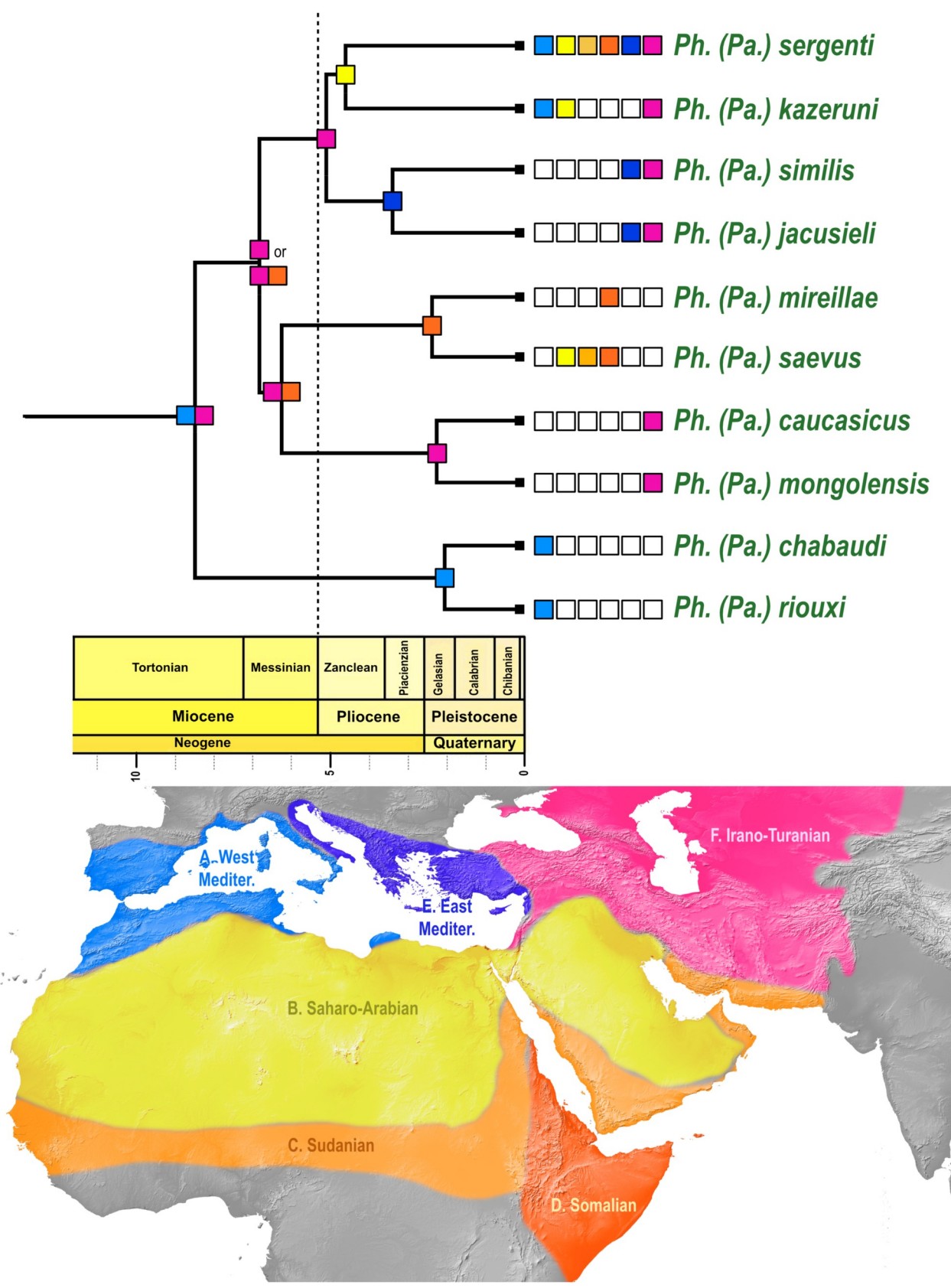

**Fig 3. Ancestral range estimation for the subgenus *Paraphlebotomus sensu nov.*** Geographic areas as delineated on the map are represented with colored boxes. A: West Mediterranean region; B: Saharo-Arabian region; C: Sudanian region; D: Somalian region; E: East Mediterranean region; F: Irano-Turanian. Most probable ancestral states (areas or combination of areas) as inferred by DEC and DIVALIKE are indicated at each node of the phylogeny. When states are different between the two models, the most probable state as inferred by DIVALIKE is indicated first. Input data (matrix of geographic ranges for species included in the analysis, time periods and dispersal multipliers); ancestral range estimations for all tested models (DEC, DIVALIKE, BAYAREALIKE), and results of model testing are available in S1–S4 Text. The background map was in the public domain and has been downloaded from Natural Earth (http://www.naturalearthdata.com).

## Systematics of the subgenus *Paraphlebotomus sensu nov*

Our phylogenomic analysis (Figs 2 and S2) recovers fully resolved relationships within subgenus *Paraphlebotomus sensu nov.* that corresponds well to morphological affinities. The first clade formed by *Ph. chabaudi* and *Ph. riouxi* is supported by several morphological characters. Females of these North African species exhibit a typical spermathecae with a bellflower-like distal ring and males differ from all other species by their parameral sheaths that are apically pointed. *Ph. chabaudi* and *Ph. riouxi* can be differiented by the size of the basal lobes of their gonocoxites and the number of setae they harbor. Based on dubious molecular evidences [and despite the tree presented in Fig 4 of their 2014 paper shows that specimens of the two species belong to two different clades], Tabbabi et al. [62,63] considered these species as synonyms. This result was acknowledged by Grace-Lema et al. [20], who wrongly interpreted a highly supported sister-taxa relationship between *Ph. chabaudi* and *Ph. riouxi* as a support for their synonymy, although they noticed the long branch between the two species. The high genetic distance is mis-interpreted by the author as a consequence of missing data, which is an inconsistent statement as other analysed species show the same level of divergence but are not considered as synonyms. The present study confirms the validity of *Ph. riouxi*, that was already assessed with Sanger data and morphological evidence by Lehrter et al. [64]. Therefore *Ph. riouxi* stat. rev. is reinstated as a valid species, from its status of junior synonym of *Ph. chabaudi*.

The clade grouping all other species of *Paraphlebotomus* is subdivided into four pairs of sister taxa that exhibit strong morphological affinities:

1. Females of the Afrotropical *Ph. saevus* and *Ph. mireillae* mainly differ by the presence of spines on spermathecal ducts. Males, which are extremely close, differ by the distal part of the parameral sheath (thicker in *Ph. saevus* than in *Ph. mireillae)*, and the hairs of the gonocoxite basal lobe (longer in *Ph. mireillae* than in *Ph. saevus)*. These two species are morphologically closely related to *Ph. gemetchi* and *Ph. nuri*, two rare species occurring respectively in Ethiopia and Pakistan, that we could not include in the present study.

2. Males of *Ph. caucasicus* and *Ph. mongolensis* are closely related but recognizable by the shape of gonostyle or of basal lobe of the gonocoxite. To the contrary, females are undistinguishable and share mitochondria [65], a putative introgression that supports the close relationships observed in our topologies. *Ph. andrejevi*, another species that could not be included in our analysis certainly belongs to this clade as females of *Ph. andrejevi* are impossible to discriminate from *Ph. caucasicus* and *Ph. mongolensis*.

3. Males of *Ph. similis* and *Ph. jacusieli* share many characters (shape of the parameral sheath or the flagellomere lengths) but differ by the longer process carrying the most distal spine of the gonostyle in *Ph. jacusieli* and the thick pharyngeal teeth of the female (occupying the whole pharynx in *Ph. similis* versus restricted to the central and anterior part of the armature in *Ph. jacusieli*).

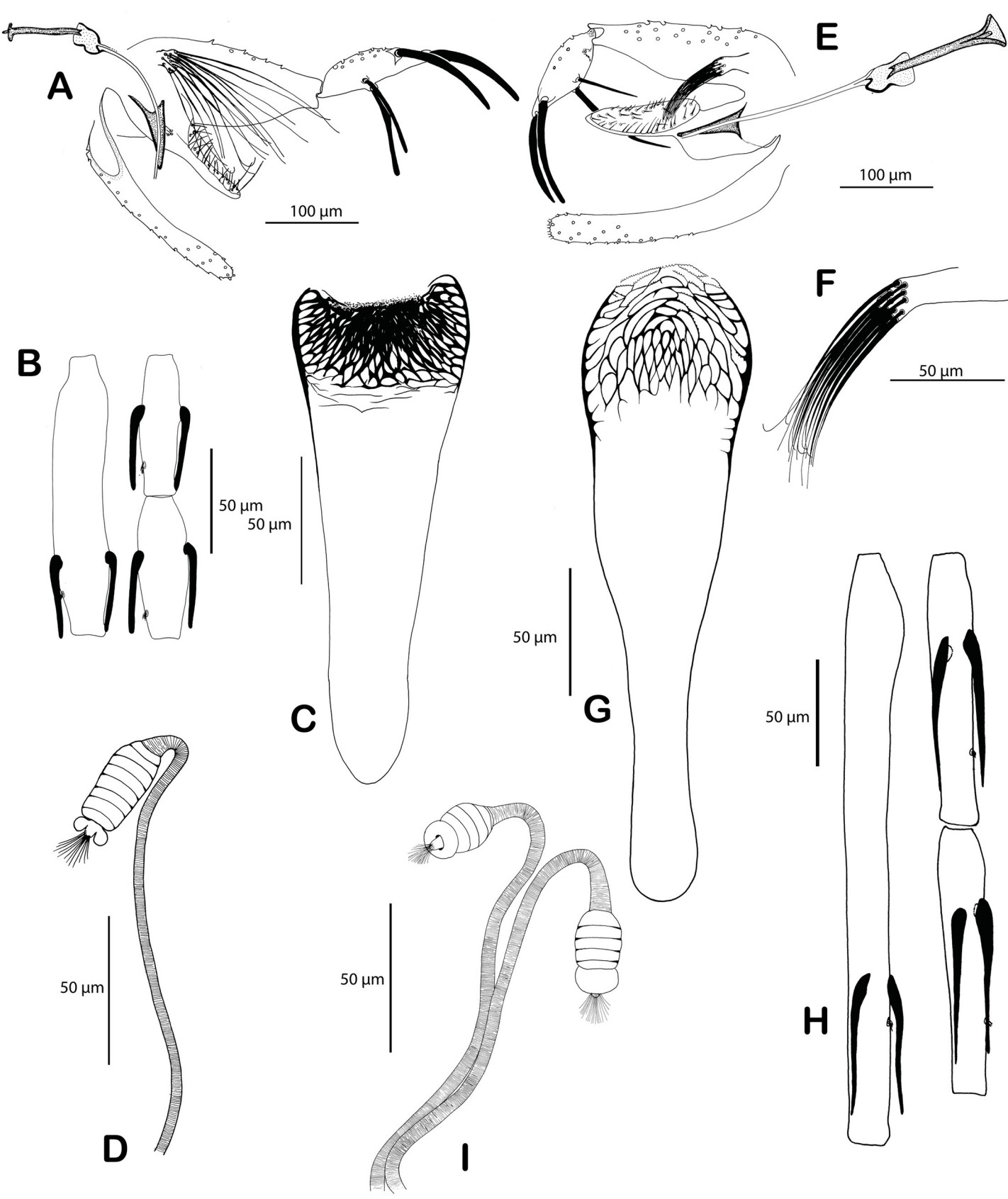

**Fig 4. Comparison of *Ph. alexandri*, the type-species of *Artemievus subg. nov.* and *Ph. sergenti*, the type-species of the subgenus *Paraphlebotomus*.** *Ph. alexandri*: genitalia of the lectotype from Amara, Iran (A), female from Syria: flagellomeres 1, 2, 3 (B), pharynx (C) and spermatheca (D). *Ph. sergenti* syntypes from Mac Mahon, Algeria: male genitalia (E) and detail of the basal lobe of the coxite (F); female pharynx (G), flagellomeres 1, 2, 3 (H) and spermathecae (I). Drawings by J. Depaquit.

4. Males of *Ph. sergenti* and *Ph. kazeruni* share a short and thin curved basal lobe of the coxite and similar gonostyle, whereas females share a similar pharyngeal armature. Females of *Ph. kazeruni* differs from those of *Ph. sergenti* by the lowest number of spermathecal rings (only one or two). *Ph. sergenti* + *Ph. kazeruni* cluster with *Ph. similis* + *Ph. jacusieli* in a strongly supported clade, in agreement with their morphological affinities (shape of the parameral sheath, flagellomeres lengths, size of the basal lobe of the gonocoxite).

## Evolutionary history and biogeography of *Paraphlebotomus sensu nov*

From the Mid-Miocene Climate Optimum (16.4–14.7 Ma) to present time, the Mediterranean region and the adjacent Middle Eastern areas experienced multiple geologic and climatic events that have strongly affected the distribution of organisms. Our analyses indicate that the most ancient ancestor of present *Paraphlebotomus* species originated ca 12.9 Ma at a time of global cooling and strong environmental changes (Fig 2). DEC and DIVALIKE models indicate that the range of the MRCA is ambiguous and suggest that the most likely range is composed of two disjoint areas: West-Mediterranean + Irano-Turania, which seems implausible (Fig 3 and S1 Text). This result can be partly explained by the fact that i) some lineages were lost; ii) distribution data are still patchy; iii) biogeographic studies for the subgenera *Phlebotomus* and *Synphlebotomus* are lacking and it was not possible to estimate ancestral range for

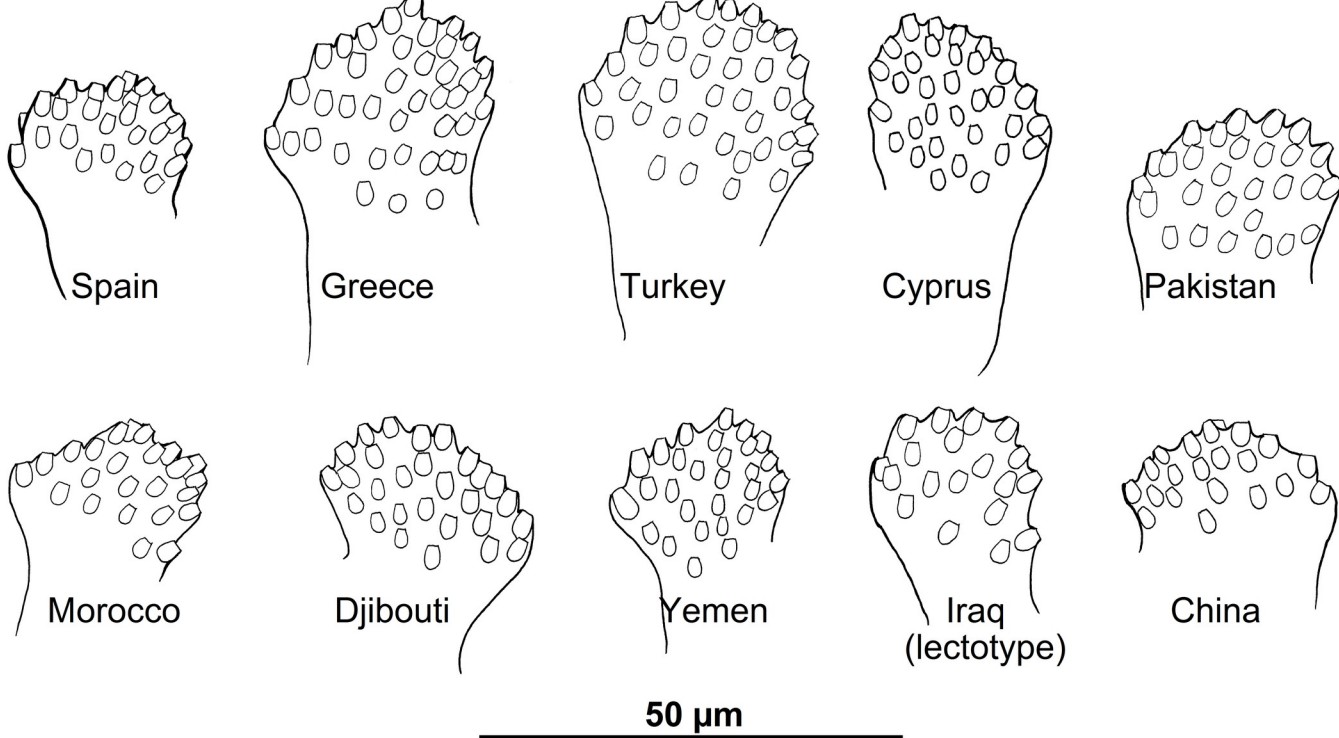

Spain Greece Turkey Cyprus Pakistan

Morocco Djibouti Yemen Iraq (lectotype) China

50 µm

**Fig 5. Morphological variability of the basal lobe of the gonocoxite of *Ph. alexandri* specimens collected in different localities.** Drawings by J. Depaquit.

these two subgenera in this study because very few species were included. Two different hypotheses could be considered for the MRCA of *Paraphlebotomus*: a restricted distribution or a large distribution (like *Ph. sergenti* or other phlebotomine sandflies). DEC and DIVALIKE appear to favor a largely distributed MRCA which is compatible with paleogeographical and paleoclimatic events.

During the Early and Middle Miocene, closure of the gateway of Proto-Mediterranean Sea [66] followed by the Messinian Salinity Crisis [67] and coupled with the uplift of the Atlas and Zagros Mountains [68], have strongly affected climatic conditions in the area under consideration. These mountain ranges, as well as the uplift of the Qinghai-Tibetan Plateau on the East, have acted as topographic barriers to moisture and have largely contributed to the aridification of the area. As a consequence, the vegetation of the area dominated by forests in the Early Miocene, underwent a transition to open grassland vegetation towards the late Miocene. Grasslands may have favored the expansion of the most ancient common ancestor of present *Paraphlebotomus*, enabling it to range from the Maghrebian mountains to central Asia.

The expansion of the Sahara as an arid belt has long been dated back to the Quaternary Ice age (ca 2.5 Ma) [69] or to the Pliocene (5.33–2.58 Ma) [70,71]. Recent studies [72,73] have demonstrated that arid desert conditions existed repetitively since 7 Ma (Late Miocene). Zhang et al [74] used climate model simulations and confirmed that the Late Miocene (ca 7–11 Ma) has been the pivotal period for triggering north African aridity and creating the Sahara Desert. An early origin of the Sahara Desert is also suggested by the work of Böhme et al [75] who dated a saline aeolian dust deposit from Sahara in Greece back to 7.37–7.11 Ma. Therefore, the basal split between the Maghrebian clade (*Ph. riouxi* + *Ph. chabaudi*) and other *Paraphlebotomus* species, ca 8.5 Myr ago, may have resulted from the rise of the Saharan belt and the aridification of the region located between the Atlas Mountains and the Turanian subregion.

During the Messinian, the Mediterranean region and the Middle East underwent rapid and dramatic climatic changes. Multiple factors probably operated simultaneously, the expansion of Antarctic ice-sheet [76], the rise of Sahara [77], the closure of the connection between the Atlantic Ocean and the Mediterranean sea that leaded to the nearly complete desiccation of the Mediterranean sea (Messinian salinity crisis) [67]. Consequently, steppic vegetation [78] and desert areas [79] expanded in the sub-desert belt south of Mediterranean region and extending to Middle and Central Asia. Therefore, the disjunct occurrence of the ancestors of the four lineages present in the East Mediterranean (*Ph. similis* + *Ph. jacusieli*), Saharo-Arabian (*Ph. sergenti* + *Ph. kazeruni*), Somalian (*Ph. mireillae* + *Ph. saevus*, possibly also *Ph. gemetchi*) and Turanian subregions (*Ph. caucasicus* + *Ph. mongolensis*, possibly also *Ph. nuri* and *Ph. andrejevi*) is probably the result of a Messinian vicariant event in which the extreme prolonged drought has been instrumental.

Our biogeographical analyses dated the last episode of speciation within the subgenus *Paraphlebotomus* during the Pleistocene (between 3 and 2 Ma). At that time, the Middle East areas and the East African subregion have experienced cycles of wet periods followed by periods of strong droughness [80]. Interestingly, these cycles have also favored the diversification of the rodents, in the burrows of which *Paraphlebotomus* larvae develop [81–83]. In arid natural areas, the presence of sand fly species is strongly limited by the low moisture of the soil [84]. *Paraphlebotomus* species therefore occur close to sites where larvae can develop. Rodent burrows maintain greater humidity and *Paraphlebotomus* larvae can develop upon organic matter (feces) that accumulate in the moist atmosphere of the deepest burrows [84]. Burrows also provide good shelter to adults during the warmer hours of the day. This coupled to the relatively low dispersal abilities of *Phlebotomus* species (less than 2 kilometers, [85]) and the relative scarceness of suitable larval habitats in the dryer and warmest areas, make sand fly population

scanty and prone to local extinction in desert and sub-desert habitats. Therefore, cycles of dry-wet periods may have resulted in reduction and separation of sand fly populations and finally by the formation of the current species. Subsequently, a few species have experienced large expansion in the last million years and have evolved toward anthropophily (i.e. *Ph. sergenti*).

## Putative species complexes that required further studies

The widely distributed species *Ph. alexandri* and *Ph. sergenti* exhibit high intra-specific divergences, which suggests that these species may represent complexes of closely related entities. Two subclades are recovered in *Ph. alexandri* (Figs 2 and S2). The genetic distance between these subclades (1.6% on 1,024,190 nt) is similar to the distance observed between other pairs of sister species present in our topology (e.g. 1.6% between *Ph. similis* and *Ph. jacusieli* and 1.7% between *Ph. chabaudi* and *Ph. riouxi*). *Ph. alexandri* exhibits a high variability in the number of setae on the basal lobe of the gonocoxite and this variability appears to be linked to the geographical origin of the studied specimens (Fig 5). In the same manner, the genetic distance observed between individuals of *Ph. sergenti* from the West Mediterranean area (Portugal, Algeria, Morocco) and those from Egypt and Afghanistan are respectively of 1.1% and 1.2%. The divergence observed within *Ph. sergenti* is slightly lower than the one observed in *Ph. alexandri* but however remains substantial. Such a high intraspecific divergence within these two species has already been observed [86–88] and deserves further studies to clarify the exact status of the subclades observed within *Ph. alexandri* and *Ph. sergenti*. This is of prime importance as it may have strong consequences on the epidemiology of *Leishmania infantum*, *L. major* and to a lesser extent of *L. donovani*. Importantly, our results highlight the power of nuclear pangenomic markers over mitochondrial genes to analyse genetic differentiation and phylogeography of phlebotomine sand flies. Indeed, several species-complexes of *Phlebotomus* may have experienced mitochondrial introgression and share haplotypes between species, even when nuclear gene flow is restricted [65,89,90]. In these complex, frequent mito-nuclear discordance due to introgression, incomplete lineage sorting, endosymbionts or mis-identification renders species delimitation difficult. This is probably where reliable markers for identification are mostly wanted but this is also where barcoding approaches (based on *COI* or *Cyt*b genes) may be deficient and have to be used cautiously.

## RADseq to improve our understanding of the epidemiology of arthropod borne diseases

RAD sequencing is now widely used for population genetics and phylogenetics. Notably, the method has been shown effective to decipher species complexes or to detect introgression [91]. However, it is still rarely used to study minute insects due to the limiting DNA amount that can be extracted. Here, we confirm that multiple displacement amplification method can be used to generate unbiased copies of genomes to make possible the sequencing of RAD markers from tiny specimens [34]. Coupled with continuous progress in high-throughput sequencing technology, ever larger numbers of samples can be processed at the same time which make possible large-scale studies. For instance, with the *PstI* enzyme used here, about 400 individuals of phlebotomines could be sequenced on a S1 NovaSeq flow cell.

Several studies have shown that members of phlebotomine species complexes can display contrasted differences in terms of biology and vector capacity [92]. Therefore, deciphering species-complexes with such a high-throughput and powerful method may contribute to better identify biological entities of importance for the transmission of *Leishmania*. Further, RADseq characterization of species complexes may help to i) better delimit the geographical distribution of the different species; ii) identify phenotypic traits relevant to the disease epidemiology

and iii) analyse intraspecific population structure within entities and genetic exchanges, if any, between them.

Once thoroughly delimitated molecularly, the specific level of these entities could be validated by crossbreeding experiments and/or other analyses focusing on chemical and behavioral characters (partner courtship, attraction by sex pheromones, etc). Subsequently, species need to be characterized morphologically, if possible, and named. The accurate species delimitation done using WGA and RAD markers, coupled with non-destructive extraction (see [93]), may facilitate the identification of diagnostic morphological characters. Finally, "barcoding-like" markers that enable accurate molecular identification of the species within analysed complexes could then be developed to enable fast and massive identification to survey large areas. The WGA+RAD approach developed here, that can be used on single specimens, may therefore represent a powerful tool to improve our understanding of the epidemiology of leishmaniases and possibly also other arthropod borne human diseases.

## Supporting information

**S1 Text. Input data and results of the ancestral range estimation using BioGeoBears.** Matrix of geographic ranges and map of geographic regions
(DOCX)

**S2 Text. Input data and results of the ancestral range estimation using BioGeoBears.** Dispersal multipliers matrices and Time Periods
(DOCX)

**S3 Text. Input data and results of the ancestral range estimation using BioGeoBears.** Results of model testing
(DOCX)

**S4 Text. Input data and results of the ancestral range estimation using BioGeoBears.** Ancestral range estimations for all tested models as inferred by BioGeoBears
(DOCX)

**S1 Fig. Phylogenetic tree obtained for the *COI* data set**
(PNG)

**S2 Fig. Phylogenetic tree obtained for the RAD data sets.** cstacks n = 6; radis_nsample_min = 50%; 5,050 loci: RAxML tree (unpartitioned data set); Bootstrap values at nodes (100 replicates); %missing = % of missing RAD tags. Data sets are described in Table 2.
(BMP)

**S3 Fig. Phylogenetic tree obtained for the RAD data sets.** cstacks n = 6; radis_nsample_min = 75%; 1,423 loci; RAxML tree (unpartitioned data set); Bootstrap values at nodes (100 replicates); %missing = % of missing RAD tagsData sets are described in Table 2.
(BMP)

**S4 Fig. Phylogenetic tree obtained for the RAD data sets.** cstacks n = 8; radis_nsample_min = 50%; 7,194 loci; RAxML tree (unpartitioned data set); Bootstrap values at nodes (100 replicates); %missing = % of missing RAD tags. Data sets are described in Table 2.
(BMP)

**S5 Fig. Phylogenetic tree obtained for the RAD data sets.** cstacks n = 8; radis_nsample_min = 75%; 2,086 loci; RAxML tree (unpartitioned data set); Bootstrap values at nodes (100 replicates); %missing = % of missing RAD tags. Data sets are described in Table 2.
(BMP)

**S6 Fig. Phylogenetic tree obtained for the RAD data sets.** cstacks n = 10; radis_nsample_-min = 50%; 8,906 loci; RAxML tree (unpartitioned data set); Bootstrap values at nodes (100 replicates); %missing = % of missing RAD tagsData sets are described in Table 2.
(BMP)

**S7 Fig. Phylogenetic tree obtained for the RAD data sets.** cstacks n = 10; radis_nsample_-min = 75%; 2,808 loci; RAxML tree (unpartitioned data set); Bootstrap values at nodes (100 replicates); %missing = % of missing RAD tags Data sets are described in Table 2.
(BMP)

**S1 Table. Information about samples used in this study and results of the RAD-seq experiment.**
(XLSX)

## Acknowledgments

The authors thank Carlos Alves-Pires, Azzedine Bounamous, Vit Dvořák, Nabil Haddad, Hanafi Hanafi, Najoua Haouas, Vladimir Ivovic, Kaouther Jaouadi, Mireille and Bob[†] Killick-Kendrick, Andreas Krüger, Nicole Léger, Michele Maroli, Laor Oshan, Yusuf Özbel, Jean-Antoine Rioux[†], and Petr Volf for providing specimens and/or for fruitful discussions about the distribution of *Ph. similis* and *Ph. sergenti* in Turkey and in the Balkans. We thank Zoe Adams (NHM, London) and Zoubir Harrat (Pasteur Institute of Algeria) who arranged the loans of type-specimens of *Ph. alexandri* and *Ph. sergenti*, respectively. We also are grateful to Morad Belbagra for the technical assistance in data processing, to Laure Sauné for advices during library preparation, to Montpellier GenomiX (MGX, France) for sequencing and to the genotoul bioinformatics platform Toulouse Occitanie (Bioinfo Genotoul, doi: 10.15454/1.5572369328961167E12) for providing computing resources.

## Author Contributions

**Conceptualization:** Jean-Yves Rasplus, Jérôme Depaquit.

**Data curation:** Astrid Cruaud, Véronique Lehrter, Guenaëlle Genson.

**Formal analysis:** Astrid Cruaud, Véronique Lehrter, Guenaëlle Genson, Jean-Yves Rasplus, Jérôme Depaquit.

**Funding acquisition:** Jean-Yves Rasplus, Jérôme Depaquit.

**Investigation:** Astrid Cruaud, Véronique Lehrter, Guenaëlle Genson, Jean-Yves Rasplus.

**Methodology:** Astrid Cruaud, Véronique Lehrter, Guenaëlle Genson, Jean-Yves Rasplus, Jérôme Depaquit.

**Resources:** Jérôme Depaquit.

**Supervision:** Astrid Cruaud, Véronique Lehrter, Jean-Yves Rasplus, Jérôme Depaquit.

**Validation:** Astrid Cruaud, Véronique Lehrter, Guenaëlle Genson, Jean-Yves Rasplus, Jérôme Depaquit.

**Visualization:** Astrid Cruaud, Véronique Lehrter, Guenaëlle Genson, Jean-Yves Rasplus, Jérôme Depaquit.

**Writing – original draft:** Astrid Cruaud, Jean-Yves Rasplus.

**Writing – review & editing:** Astrid Cruaud, Véronique Lehrter, Guenaëlle Genson, Jérôme Depaquit.

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
