## [Decision Letter · Decision Letter 0]

15 Apr 2021

Dear Dr. Depaquit,

Thank you very much for submitting your manuscript "Evolution, systematics and historical biogeography of sand flies of the subgenus Paraphlebotomus (Psychodidae) inferred using restriction-site associated DNA markers." for consideration at PLOS Neglected Tropical Diseases. As with all papers reviewed by the journal, your manuscript was reviewed by members of the editorial board and by several independent reviewers. The reviewers appreciated the attention to an important topic. Based on the reviews, we are likely to accept this manuscript for publication, providing that you modify the manuscript according to the review recommendations. 

Sincerely,

Remi N. Charrel

Associate Editor

Jaap van Hellemond

Deputy Editor

Reviewer's Responses to Questions

**Key Review Criteria Required for Acceptance?**

**Methods**

-Are the objectives of the study clearly articulated with a clear testable hypothesis stated?

-Is the study design appropriate to address the stated objectives?

-Is the population clearly described and appropriate for the hypothesis being tested?

-Is the sample size sufficient to ensure adequate power to address the hypothesis being tested?

-Were correct statistical analysis used to support conclusions?

-Are there concerns about ethical or regulatory requirements being met?

Reviewer #1: Are the objectives of the study clearly articulated with a clear testable hypothesis stated?

Yes.

Hypotheses - With the use of advanced molecular techniques that make it possible to obtain impartial copies of the entire genome when there is a low amount of DNA and obtain sequencing of pangenomic markers for tiny species, as is the case of sandflies to be used in phylogenetic analysis with results robust?

The objectives - To investigate for the subgenus Paraphlebotomus (Diptera, Psychodidae, Phlebotominae), constituted by important vector species of Leishmania that cause visceral and cutaneous leishmaniasis in the Old World, the phylogeny of the group, as well as divergence time estimates and probabilistic inference of historical biogeography of the subgenus.

-Is the study design appropriate to address the stated objectives?

Yes

The design of the study involved obtaining the sequences (Whole Genome Amplification, WGA) and RAD (Restriction site Associated DNA) of most species of the subgenus to develop the phylogenetic analysis and also of specimens from most of the areas where they occur. Therefore, I consider the design appropriate to meet the objectives.

-Is the population clearly described and appropriate for the hypothesis being tested?

Studies focusing on the systematics of groups generally use few individuals. In the present case, this is a great differential, since from the genomic information of a few individuals it was possible to obtain robust inferences about the phylogeny of the group. The information for the specimens is adequate.

-Is the sample size sufficient to ensure adequate power to address the hypothesis being tested?

See comments above

-Were correct statistical analysis used to support conclusions?

Yes

The statistics used to infer phylogeny have often been adopted in studies of phylogenetic systematics. To estimate the time of divergence between the clades they adopted a recently proposed method - RelTime method, relative rate framework, which presents results very close to those of more conventional methods, for example Bayesian framework.

-Are there concerns about ethical or regulatory requirements being met

Not applicable

Reviewer #2: The objectives are clear, the study is well design.

I just have a comment: L268-L271. I do not think this part is necessary.

**Results**

-Does the analysis presented match the analysis plan?

-Are the results clearly and completely presented?

-Are the figures (Tables, Images) of sufficient quality for clarity?

Reviewer #1: -Does the analysis presented match the analysis plan?

The analysis plan was presented and commented on for each topic.

-Are the results clearly and completely presented?

Phylogenetic trees are illustrated and commented using COI which, like other studies of the marker, does not present good phylogeny resolution. On the other hand, trees with RAD sequences considering the parameters that could limit the results confirm the same clades for all sets of parameters, with great robustness. On the other hand, the trees clearly show the paraphilia of one of the species that until then comprised the subgenus Paraphlebotomus and for which a new subgenus has been proposed. The trees that show the clades' divergence time as well as the probable locations that each clade appears according to the regions of the Old World are illustrated and commented.,

-Are the figures (Tables, Images) of sufficient quality for clarity?

With the exception of Table S1, which was not available, all tables and images are of high quality.

Reviewer #2: The analysis presented match the analysis plan, they are clear and completely presented.

The figures are good and clear. 

I just have a question, L293: “16.6–34.6% missing data”. Do the authors have any idea why this rate? Is it really low 1/3 of missing data?

**Conclusions**

-Are the conclusions supported by the data presented?

-Are the limitations of analysis clearly described?

-Do the authors discuss how these data can be helpful to advance our understanding of the topic under study?

-Is public health relevance addressed?

Reviewer #1: -Are the conclusions supported by the data presented?

Yes. 

-Are the limitations of analysis clearly described?

Yes 

-Do the authors discuss how these data can be helpful to advance our understanding of the topic under study?

The methodological approach used to infer phylogeny is discussed and, most likely, it will generate an impact among researchers for systematic studies in obtaining phylogenies, defining more faithfully the monophyletic groups and the possibility of retrieving the evolutionary history of the groups.

-Is public health relevance addressed?

The relevance of the study was commented by the authors, in the sense that the group comprises several vector species of Leishmania that cause visceral and cutaneous leishmaniasis in the Old World. Knowing the relationship of the focus group with its sister groups and between species within the group, can also help to understand the agent-vector-host relationship.

Reviewer #2: The article is well written, interesting and complete.

It could be a plus to have pictures of the described specimens especially for people having less notion in identification of sandflies. Example L 349 « Artemievus subg. nov. Depaquit. Type-species: Phlebotomus alexandri Sinton, 1928. »

**Editorial and Data Presentation Modifications?**

Reviewer #1: Minor corrections to be made.

 –line 27 Ph. alexandri, replace by Phlebotomus alexandri. 

Line 32 – give space after 5.3

Line 142 – Nine outgroups replace with Nine outgroup specimens

Lines 163 to 177 – replace µl with µL 

line 331 and (italic); it shoud be non italic

In table 1. Standardize the number of letters for the subgenera. Only Paraphlebotomus has four letters. 

Make table S1 available.

Reviewer #2: (No Response)

**Summary and General Comments**

Reviewer #1: Excellente work. As suggestion regarding the new subgenus proposed, I think it would add quality to the work if illustrations were presented on the morphological characters that make the distinction between Paraphlebotomus and Artemievus possible. I suggest illustrations of the type-species of Paraphlebotomus and Phlebotomus alexandri.

Reviewer #2: As a summary comments, I can only say that it is a good article.

PLOS authors have the option to publish the peer review history of their article (what does this mean?). If published, this will include your full peer review and any attached files.

Reviewer #1: No

Reviewer #2: No

Figure Files:

Data Requirements:

Reproducibility:

References

---

## [Editor Report · Decision Letter 1]

15 May 2021

Dear Dr. Depaquit,

We are pleased to inform you that your manuscript 'Evolution, systematics and historical biogeography of sand flies of the subgenus Paraphlebotomus (Psychodidae) inferred using restriction-site associated DNA markers.' has been provisionally accepted for publication in PLOS Neglected Tropical Diseases.

Best regards,

Remi N. Charrel

Associate Editor

Jaap van Hellemond

Deputy Editor

---

## [Editor Report · Acceptance letter]

13 Jul 2021

Dear Dr. Depaquit,

We are delighted to inform you that your manuscript, "Evolution, systematics and historical biogeography of sand flies of the subgenus Paraphlebotomus (Diptera, Psychodidae, Phlebotomus) inferred using restriction-site associated DNA markers," has been formally accepted for publication in PLOS Neglected Tropical Diseases.

Best regards,

Shaden Kamhawi

co-Editor-in-Chief

Paul Brindley

co-Editor-in-Chief
